# Feasibility, Safety and Impact on Overall Survival of Awake Resection for Newly Diagnosed Supratentorial *IDH*-Wildtype Glioblastomas in Adults

**DOI:** 10.3390/cancers13122911

**Published:** 2021-06-10

**Authors:** Alessandro Moiraghi, Alexandre Roux, Sophie Peeters, Jean-Baptiste Pelletier, Marwan Baroud, Bénédicte Trancart, Catherine Oppenheim, Emmanuèle Lechapt, Chiara Benevello, Eduardo Parraga, Pascale Varlet, Fabrice Chrétien, Edouard Dezamis, Marc Zanello, Johan Pallud

**Affiliations:** 1Department of Neurosurgery, GHU Site Sainte-Anne, F-75014 Paris, France; alessandro.moiraghi@sfits.ch (A.M.); a.roux@ghu-paris.fr (A.R.); b.trancart@ghu-paris.fr (B.T.); c.oppenheim@ghu-paris.fr (C.O.); c.benevello@ghu-paris.fr (C.B.); e.parraga@ghu-paris.fr (E.P.); e.dezamis@ghu-paris.fr (E.D.); m.zanello@ghu-paris.fr (M.Z.); 2Université de Paris, 102-108 rue de la Santé, F-75014 Paris, France; e.lechapt@ghu-paris.fr (E.L.); p.varlet@ghu-paris.fr (P.V.); f.chretien@ghu-paris.fr (F.C.); 3Division of Neurosurgery, Geneva University Hospitals and Faculty of Medicine, University of Geneva, 1211 Geneva, Switzerland; 4Swiss Foundation for Innovation and Training in Surgery (SFITS), 1205 Geneva, Switzerland; 5Institut de Psychiatrie et Neurosciences de Paris (IPNP), UMR S1266, INSERM, IMA-BRAIN, 75014 Paris, France; 6Department of Neurosurgery, University of California, Los Angeles, CA 90095, USA; speeters@mednet.ucla.edu; 7Department of Neurosurgery, CHU de Saint Etienne, 42270 Saint Etienne, France; jean-baptiste.pelletier@neurochirurgie.fr; 8Department of Neurosurgery, CHU Notre Dame de Secours, Byblos, Lebanon; dm@chu-nds.org; 9Department of Neuroradiology, GHU Site Sainte-Anne, F-75014 Paris, France; 10Department of Neuropathology, GHU Site Sainte-Anne, F-75014 Paris, France

**Keywords:** awake surgery, glioblastoma, *IDH*-wildtype, extent of resection, survival

## Abstract

**Simple Summary:**

A few studies have suggested the benefits of awake surgery by maximizing the extent of resection while preserving neurological function and improving survival in high-grade glioma patients. However, the histomolecular heterogeneity in these series, mixing grade 3 with grade 4, and *IDH*-mutated with *IDH*-wildtype gliomas, represents a major selection bias that may influence survival analyses. For the first time, in a large homogeneous single-institution cohort of newly diagnosed supratentorial *IDH*-wildtype glioblastoma in adult patients, we assessed feasibility, safety and efficacy of awake surgery using univariate, multivariate and case-matched analysis. Awake surgery was associated with higher resection rates, lower residual tumor rates, and more supratotal resections than asleep resections, allowed standard radiochemotherapy to be performed systematically within a short time between surgery and radiotherapy, and was an independent predictor of progression-free survival and overall survival in the whole series, together with the extent of resection, *MGMT* promoter methylation status, and standard.

**Abstract:**

Background: Although awake resection using intraoperative cortico-subcortical functional brain mapping is the benchmark technique for diffuse gliomas within eloquent brain areas, it is still rarely proposed for IDH-wildtype glioblastomas. We have assessed the feasibility, safety, and efficacy of awake resection for IDH-wildtype glioblastomas. Methods: Observational single-institution cohort (2012–2018) of 453 adult patients harboring supratentorial IDH-wildtype glioblastomas who benefited from awake resection, from asleep resection, or from a biopsy. Case matching (1:1) criteria between the awake group and asleep group: gender, age, RTOG-RPA class, tumor side, location and volume and neurosurgeon experience. Results: In patients in the awake resection subgroup (*n* = 42), supratotal resections were more frequent (21.4% vs. 3.1%, *p* < 0.0001) while partial resections were less frequent (21.4% vs. 40.1%, *p* < 0.0001) compared to the asleep (*n* = 222) resection subgroup. In multivariable analyses, postoperative standard radiochemistry (aHR = 0.04, *p* < 0.0001), supratotal resection (aHR = 0.27, *p* = 0.0021), total resection (aHR = 0.43, *p* < 0.0001), KPS score > 70 (HR = 0.66, *p* = 0.0013), *MGMT* promoter methylation (HR = 0.55, *p* = 0.0031), and awake surgery (HR = 0.54, *p* = 0.0156) were independent predictors of overall survival. After case matching, a longer overall survival was found for awake resection (HR = 0.47, *p* = 0.0103). Conclusions: Awake resection is safe, allows larger resections than asleep surgery, and positively impacts overall survival of IDH-wildtype glioblastoma in selected adult patients.

## 1. Introduction

*Isocitrate Dehydrogenase* (*IDH*)-wildtype glioblastomas are the most common malignant primary brain tumor in adults [1,2,3]. *IDH*-wildtype glioblastoma survival is linked to the extent of resection and to the patients’ functional status, since postoperative neurological and cognitive decline impacts patients’ recovery and subsequently timely access to adjuvant radiochemotherapy [3,4,5,6,7,8]. The goals for surgery are maximizing the extent of resection while minimizing the morbidity [9,10,11,12]. For that reason, *IDH*-wildtype glioblastomas located in eloquent brain areas are usually not resected as aggressively as glioblastomas not involving eloquent brain areas, due to the risks of permanent severe neurologic damage reducing life expectancy [6,13]. To improve the extent of resection, several intraoperative techniques have been proposed. 5-ALA fluorescence has been proven effective for improving the overall survival by a randomized multicenter clinical trial [10,14]. Sodium fluorescein fluorescence also showed good results in terms of extent of resection, but no significant results on overall survival have been reported [15]. Similarly, intraoperative MRI showed an impact on glioblastoma resection and patient survival [11], while intraoperative ultrasound was found useful in performing more radical surgeries and preventing neurological impairment, but large series reporting a benefit on survival are still lacking [9,16]. To improve the safety of the surgical resection and to improve its benefit-to-risk ratio, awake surgery is the benchmark intraoperative technique for gliomas in eloquent brain areas [17,18,19]. While still rarely proposed for *IDH*-wildtype glioblastomas, a few studies have suggested the benefits of awake surgery by maximizing the extent of resection while preserving neurological function and improving survival in high-grade glioma patients [20,21,22,23,24]. However, the histomolecular heterogeneity in these series, mixing grade 3 with grade 4, and *IDH*-mutated with *IDH*-wildtype gliomas, represents a major selection bias that may influence survival analyses.

We report the neurosurgical management of newly diagnosed supratentorial *IDH*-wildtype glioblastomas in a homogeneous single-institution cohort of adult patients. We have assessed the feasibility and safety (intraoperative findings, postoperative complications, and outcomes), and efficacy (access to adjuvant radiochemotherapy, progression-free survival, and overall survival) of awake surgery by comparing it to asleep surgery using case matching, and by stratifying according to the neurosurgeon’s experience.

## 2. Materials and Methods

### 2.1. Study Design

An observational retrospective cohort study was conducted at a tertiary referral neurosurgical oncology center on glioma patients between January 2012 and December 2018. The manuscript was written according to the Strengthening the Reporting of Observational Studies in Epidemiology checklist.

### 2.2. Participants

Inclusion criteria were: (1) patients ≥18 years; (2) newly diagnosed tumor; (3) supratentorial tumor location; (4) histomolecular diagnosis of *IDH*-wildtype glioblastoma according to the 2016 WHO classification and cIMPACT-NOW update 3 with histopathological re-assessment for all cases diagnosed prior to 2018 [25,26]; (5) available pre- and postoperative MRI to quantify the extent of resection; (6) no inclusion in a clinical trial to exclude any particular therapy; and (7) available postoperative follow-up.

### 2.3. Variables and Data Sources

Patient-, surgery-, and tumor-related characteristics included: sex, age, clinical sign(s), Karnofsky Performance Status (KPS) score, revised Radiation Therapy Oncology Group - Recursive Partitioning Analysis (RTOG-RPA) classes, tumor location, tumor volume, type of surgery (biopsy, asleep surgery, awake surgery), neurosurgeon experience (expert trained for both awake and asleep surgery, expert trained only in asleep surgery, general neurosurgeon), intraoperative adverse events, current intensity for intraoperative mapping, extent of resection, histomolecular diagnosis, early postoperative outcomes (seizure control, neurological status, KPS score), early postoperative complications, *O6-methylguanine-DNA methyltransferase* (*MGMT*) promoter methylation, oncological treatments, progression-free survival, and overall survival. All patients who underwent a surgical resection were classified according to Molinaro’s survival risk subgroups (1 to 4) [3].

### 2.4. Extent of Resection

The tumor volume (cm^3^) was calculated using manual segmentation of abnormal signal on post-contrast T1-weighted sequence for contrast enhancing lesions and on Fluid Attenuated Inversion Recovery (FLAIR) sequence on both pre- and postoperative MRI scans by three blinded investigators (AM, AR, and JP) for every tumor. The extent of resection was quantified using an early postoperative MRI (within 48 h) and performing manual segmentation of residual enhancing tumor, residual FLAIR hyperintensity and the surgical cavity on post-contrast volumetric T1-weighted and FLAIR sequences by the same three blinded investigators. Diffusion and perfusion sequences, when available, were systematically reviewed to check for postoperative ischemic and hemorrhagic events, and to define the presence of potential residual tumor. A total resection corresponded to the complete absence of the enhancing signal on post-contrast T1-weighted sequence. A supratotal resection was defined as the total absence of abnormal signal on post-contrast T1-weighted sequence plus the volume of the postoperative cavity being larger than the preoperative tumor volume, as previously defined [18,27,28,29]. All other cases were considered partial resections.

Progression-free survival was measured from the date of surgery to the date of evidence of progression or death. Tumor progression was defined according to the Response Assessment in Neuro-Oncology criteria in use at the time of management [30]. Overall survival was measured from the date of surgery to the date of death from any cause. Surviving patients were censored at the date of last follow-up.

### 2.5. Surgical Procedures

The decision as to whether to perform a particular surgical procedure was decided on an individual basis by the treating senior neurosurgeon according to his own surgical preferences. For cases amendable to surgery and located within eloquent brain areas, two of our institutional neurosurgeons favored ultrasonography-guided awake resections with intraoperative functional brain mapping through cortico-subcortical direct electrical stimulations using our in-house “asleep-awake-asleep” protocol previously detailed [19,29,31,32,33] (awake resection subgroup). These two neurosurgeons performed also asleep surgeries (expertly trained for both awake and asleep surgery). The other neurosurgeons performed only MRI-based neuronavigation-guided (BrainLab, Munich, Germany) resections without intraoperative functional brain mapping under asleep conditions independent of tumor location and were divided basing on their experience (expertly trained only in asleep surgery versus general neurosurgeon). The decision of whether to perform an awake or asleep resection was based solely on the treating neurosurgeon’s technical preference and comfort level, not on tumor volume, anatomical extent or location, side, laterality or dominance, or neuro-cognitive findings. Fluorescence techniques, fMRI and DTI were not used intraoperatively in this series. All cases deemed too risky for tumor resection received stereotactic biopsies under general anesthesia, as previously detailed (biopsy subgroup) [34,35].

### 2.6. Statistical Analyses

To assess the survival benefit of awake resection using intraoperative cortico-subcortical mapping in *IDH*-wildtype glioblastoma patients, we performed a case-matched analysis (1:1) with a control group of *IDH*-wildtype glioblastoma patients who underwent asleep resection without intraoperative functional brain mapping. Patients who received a biopsy only were excluded of the case matching. Each patient in the awake resection subgroup was individually matched with a control patient of the asleep resection subgroup according to the following criteria: (1) sex; (2) age (within 10 years); (3) RTOG-RPA class (3–4 versus 5–6); (4) tumor side; (5) tumor location (same lobe); (6) preoperative volume (cutoff by median); and (7) neurosurgeon (expert trained for both awake and asleep surgery versus expert trained only in asleep surgery versus general neurosurgeon). If no controls matched in all seven criteria, then age, RTOG-RPA classes, tumor side and tumor volume were matched, choosing worse values for the awake group. Table 1 shows the characteristics of each matched pair.

Descriptive statistics were given as the mean ± standard deviation for continuous variables and as percentage for categorical variables. To compare the awake resection, asleep resection, and biopsy subgroups, univariate analyses were carried out using the chi-square or Fisher’s exact tests for comparing categorical variables, and the unpaired t-test or Mann–Whitney rank sum test for continuous variables, as appropriate. Unadjusted survival curves for progression-free survival and overall survival were plotted by the Kaplan–Meier method, using log-rank tests to assess significance for group comparison. A Cox proportional hazard model was constructed using a backward stepwise approach, adjusting for predictors previously associated at the *p* < 0.1000 level with progression or mortality in unadjusted analysis. A *p*-value < 0.0500 was considered significant. Analyses were performed using JMP 14.1.0 (SAS Institute Inc, Cary, NC, USA).

## 3. Results

### 3.1. Patient and Tumor Characteristics

A total of 453 patients were included (55.4% men, mean age 63.0 ± 12.6 years). Patient characteristics are detailed in Table 2. A resection without intraoperative functional brain mapping under general anesthesia was performed in 49.0% of cases (*n* = 222, asleep resection subgroup), an awake resection using intraoperative cortico-subcortical mapping was performed in 9.3% of cases (*n* = 42, awake resection subgroup), and a stereotactic biopsy was performed in 41.7% of cases (*n* = 189, biopsy subgroup).

Patients of the awake resection subgroup were younger (*p* < 0.0001), had a smaller tumor volume (*p* < 0.0001), were more frequently left-sided (*p* < 0.0001) and in frontal locations (*p* < 0.0001), presented less frequently with elevated intracranial pressure (*p* < 0.0001) or focal neurologic deficits (*p* = 0.0006), but more frequently with epileptic seizures (*p* < 0.0001), had a higher KPS score (*p* < 0.0001), and were more frequently RTOG-RPA classes 3-4 (*p* < 0.0001).

### 3.2. Awake Surgery Procedure

The 42 patients were cooperative and none of them required termination of the procedure. Intraoperatively, positive functional mapping was applied in all patients at both cortical and subcortical levels at a mean stimulation current intensity of 3.6 ± 1.1 mA (range, 2.0–6.0). The overall duration of the surgery was 234.2 ± 48.9 min (range 147–365), the duration of the awakening was 13.7 ± 8.4 min (range 1–50), and the duration of the awake phase was 90.6 ± 20.3 min (range 60–140). Five patients (11.9%) had a duration of awakening >30 min, and six patients (14.3%) reported postural pain during the awake phase. A focal seizure without impaired awareness occurred intraoperatively during cortical stimulation in one patient (2.4%) and resolved after cold irrigation.

Six patients (14.3%) presented intraoperatively with elevated intracranial pressure due to tumor mass effect that precluded cortical functional mapping from being performed initially. This phenomenon was associated with higher preoperative tumor volumes (47.3 versus 15.9 cm^3^, *p* = 0.0048), but not with preoperative signs of elevated intracranial pressure (*p* = 0.7273), or neurologic focal deficits on presentation (*p* = 0.3739). In these cases, ultrasound-guided cyst puncture or intralesional debulking were performed to relieve mass effect and allow further functional mapping to be performed, as illustrated in Figure 1. The patients in question were all able to pursue intraoperative tasks after mass effect alleviation.

In all cases, resection was pursued until eloquent subcortical pathways were identified, as illustrated in Figure 2.

### 3.3. Extent of Resection

In the 264 patients who underwent a surgical resection (21.6% partial resection, 33.1% total resection, 3.5% supratotal resection), the mean extent of resection was 93.1 ± 15.7% and the mean residual tumor volume was 7.1 ± 16.2 cm^3^. In the awake resection subgroup compared to the asleep resection subgroup, the extent of resection was larger (93.9 ± 18.7 versus 92.9 ± 15.1%, *p* = 0.0313), the residual tumor volume was smaller (6.3 ± 19.2 versus 7.3 ± 15.6 cm^3^, *p* = 0.0306), supratotal resections were more frequent (21.4 versus 3.1%, *p* < 0.0001), and partial resections were less frequent (21.4 versus 40.1%, *p* < 0.0001). The rates of total resections were similar in both subgroups.

### 3.4. Postoperative Outcomes

Surgical site infections were more frequent in the asleep resection subgroup than in other subgroups (*p* = 0.0330). Worsening of a focal neurological deficit was less frequent in the biopsy subgroup than in other subgroups (*p* = 0.0299). Postoperative surgical site hematomas (*p* = 0.4490) worsened epileptic seizures (*p* = 0.1358), systemic infections (*p* = 0.0975), and thromboembolic events (*p* = 0.4514) did not vary significantly between subgroups.

All patients in the awake resection subgroup received a first-line postoperative oncological treatment while supportive care was administered for some patients in the asleep resection subgroup and even more frequently for patients in the biopsy subgroup (0 versus 5.4 versus 24.9%, *p* < 0.0001). The standard radiochemotherapy protocol was more frequently delivered to patients in the awake resection subgroup (90.5 versus 74.3 versus 61.9%, *p* < 0.0001), with a shorter time between surgery and radiotherapy (4.2 ± 2.5 versus 5.9 ± 2.5 versus 5.2 ± 2.2 weeks, *p* = 0.0008) compared to the asleep resection and biopsy subgroups.

### 3.5. Survival Analysis

The 19 patients (4.2%) lost to follow-up were excluded from survival analyses. In the whole series (*n* = 434), the median duration of postoperative follow-up was 12.0 months [95% CI, 10.0–13.5]. Two hundred and seventy-nine patients (64.3%) returned with disease progression and 327 patients (75.3%) died over the follow-up period. The median progression-free survival was 7.0 months [95% CI, 6.6–8.0], and the median overall survival was 13.6 months [95% CI, 12.0–16.0]. Kaplan–Meier curves are shown in Figure 3.

Unadjusted Hazard Ratios (HR) for progression-free survival in the whole series (*n* = 434) are detailed in Table 3. The median progression-free survival was longer following supratotal resection, than total resection, partial resection, and biopsy (26.0 months [95% CI, 9.0–not reached] versus 11.1 months [95% CI, 9.1–13.0] versus 7.0 months [95% CI, 5.0–8.0] versus 4.5 months [95% CI, 4.0–7.5], *p* < 0.0001). The median progression-free survival was longer in the awake resection subgroup, than in the asleep resection subgroup and the biopsy subgroup (17.0 months [95% CI, 11.1–26.0] versus 9.0 months [95% CI, 8.0–10.0] versus 4.7 months [95% CI, 4.0–6.0], *p* < 0.0001). After multiple adjustments using Cox models (Table 3), postoperative standard radiochemotherapy protocol (aHR = 0.08, [95% CI, 0.05–0.12], *p* < 0.0001), supratotal resection (aHR = 0.31, [95% CI, 0.15–0.65], *p* = 0.0019), total resection (aHR = 0.52, [95% CI, 0.40–0.68], *p* < 0.0001), and awake surgery (HR = 0.61, [95% CI, 0.40–0.93], *p* = 0.0157) were independently associated with longer progression-free survival. After case matching (*n* = 42 in both groups), a significantly longer progression-free survival was found for awake resection (HR = 0.59 [0.36–0.97], *p* = 0.0373): the median progression-free benefit was 4.0 months, with a median of 17.0 months [95% CI, 11.1–26.0] in the awake resection subgroup and 13.0 months [95% CI: 9.0–14.0] in the asleep resection subgroup.

Unadjusted Hazard Ratios (HR) for progression-free survival in the subgroup of patients operated on by a neurosurgeon expert both in awake and asleep surgery (*n* = 223) are detailed in Table 4. The median progression-free survival was longer following supratotal resection, than total resection, partial resection, and biopsy (26.0 months [95% CI, 9.0–not reached] versus 12.0 months [95% CI, 10.0–15.0] versus 9.0 months [95% CI, 5.6–12.0] versus 8.0 months [95% CI, 7.0–9.7], *p* < 0.0001). The median progression-free survival was longer in the awake resection subgroup, than in the asleep resection subgroup and in the biopsy subgroup (17.0 months [95% CI, 11.1–26.0] versus 10.0 months [95% CI, 9.0–13.0] versus 8.0 months [95% CI, 7.0–9.7], *p* = 0.0001). After multiple adjustments using Cox models (Table 4), postoperative standard radiochemotherapy protocol (aHR = 0.25, [95% CI, 0.15–0.41], *p* < 0.0001), supratotal resection (aHR = 0.35, [95% CI, 0.15–0.81], *p* = 0.0145), total resection (aHR = 0.62, [95% CI, 0.39–0.98], *p* = 0.0433), and awake surgery (HR = 0.63, [95% CI, 0.39–0.98], *p* = 0.0397) were independently associated with longer progression-free survival.

Unadjusted HR for overall survival in the whole series (*n* = 434) are detailed in Table 3. The median overall survival was longer following supratotal resection, than total resection, than partial resection, and than biopsy (36.0 months [95% CI, 23.0–not reached] versus 22.0 months [95% CI, 19.4–24.9] versus 12.0 months [95% CI, 0.7–15.0] versus 7.0 months [95% CI, 5.0–8.0], *p* < 0.0001). The median overall survival was longer in the awake resection subgroup, than in the asleep resection subgroup, and than in the biopsy subgroup (36.0 months [95% CI, 24.0–42.0] versus 17.2 months [95% CI, 15.1–19.4] versus 7.0 months [95% CI, 5.0–8.0], *p* < 0.0001). After multiple adjustments using Cox models (Table 3), postoperative standard radiochemotherapy protocol (Ahr = 0.22, [95% CI, 0.16–0.29], *p* < 0.0001), supratotal resection (aHR = 0.27, [95% CI, 0.12–0.62], *p* = 0.0021), total resection (aHR = 0.43, [95% CI, 0.32–0.57], *p* < 0.0001), KPS score > 70 (HR = 0.66, [95% CI, 0.52–0.85], *p* = 0.0013), *MGMT* promoter methylation (HR = 0.55, [95% CI, 0.37–0.82], *p* = 0.0031), and awake surgery (HR = 0.54, [95% CI, 0.33–0.89], *p* = 0.0156) were independently associated with longer overall survival. After case matching (*n* = 42 in both groups), a significantly longer overall survival was found for awake resection (HR = 0.47 [0.27–0.84], *p* = 0.0103). The median overall benefit was 15.0 months, with a median of 36.0 months [95% CI, 24.0–42.0] in the awake resection subgroup and 21.0 months [95% CI: 19.0–23.3] in the asleep resection subgroup.

The unadjusted HRs for overall survival in the subgroup of patients operated on by a neurosurgeon expert both in awake and asleep surgery (*n* = 223) are detailed in Table 4. The median overall survival was longer following supratotal resection, than total resection, partial resection, and biopsy (36.0 months [95% CI, 23.0–not reached] versus 23.2 months [95% CI, 19.0–25.5] versus 15.5 months [95% CI, 8.0–19.5] versus 7.0 months [95% CI, 4.9–9.3], *p* < 0.0001). The median overall survival was longer in the awake resection subgroup, than in the asleep resection subgroup and in the biopsy subgroup (36.0 months [95% CI, 24.0–42.0] versus 19.4 months [95% CI, 16.5–22.9] versus 7.0 months [95% CI, 4.9–9.3], *p* < 0.0001). After multiple adjustments using Cox models (Table 4), postoperative standard radiochemotherapy protocol (aHR = 0.23, [95% CI, 0.15–0.34], *p* < 0.0001), supratotal resection (aHR = 0.31, [95% CI, 0.29–0.71], *p* = 0.0098), total resection (aHR = 0.46, [95% CI, 0.29–0.71], *p* = 0.0005), KPS score > 70 (HR = 0.51, [95% CI, 0.35–0.74], *p* = 0.0003), *MGMT* promoter methylation (HR = 0.42, [95% CI, 0.23–0.75], *p* = 0.0037), and awake surgery (HR = 0.50, [95% CI, 0.29–0.85], *p* = 0.0115) were independently associated with longer overall survival.

## 4. Discussion

### 4.1. Key Results

In this homogeneous single-institution cohort of 453 adults harboring a supratentorial *IDH*-wildtype glioblastoma we showed that awake resection: (1) was feasible in highly selected patients with complication rates and neurological deficits inferior or similar to those of asleep resections; (2) was associated with higher resection rates, lower residual tumor rates, and more supratotal resections than asleep resections; (3) allowed standard radiochemotherapy to be performed systematically without increasing the time interval between surgery and radiotherapy; (4) was an independent predictor of progression-free survival and overall survival in the whole series, together with the extent of resection, *MGMT* promoter methylation status, and standard radiochemotherapy; (5) was a significant predictor of progression-free survival and overall survival in case-matched analyses; and (6) remained a significant predictor of survival in the subgroup of 223 patients operated on by the two neurosurgeons expert both in awake and asleep surgery.

### 4.2. Interpretation

Along with the known prognostic factors, including the extent of resection, *MGMT* promoter methylation status, and standard radiochemotherapy, we suggested that awake resection was an independent predictor of progression-free and overall survivals in *IDH*-wildtype glioblastoma patients, both in the whole series, after case matching, and in the subgroup of 223 patients operated on by the two neurosurgeons expert both in awake and asleep surgery. Two recent series reported no significant difference in topographical distribution between *MGMT* methylated and non-methylated *IDH*-wildtype glioblastomas [36,37]. Furthermore, Incekara et al. reported an association between the extent of resection, residual tumor volume, and overall survival in 326 patients with a newly diagnosed *IDH*-wildtype glioblastoma [38]. These studies confirm that *IDH*-wildtype glioblastomas can benefit from maximal resection independent from *MGMT* promoter methylation status, warranting further investigation to assess the impact of surgical techniques, including awake surgery, on overall survival. A few previous studies suggested the effects of awake surgery on survival in high-grade gliomas [21,23,39,40]. Recent systematic reviews reported promising results regarding the role of awake surgery and supratotal resection on survival in glioblastoma patients. Although promising, the level of evidence is low, consisting of small, highly selective, and retrospective series with potential confounding biases related to glioblastoma biology (*IDH*1/2 mutations status, *MGMT* promoter methylation status) [3,20,24,41,42]. We thus selected a homogeneous population of *IDH*-wildtype glioblastoma in adults, collected the *MGMT* promoter methylation status when available, and included patients from 2012 to 2018 to ensure that they were treated during the current era with similar surgical, anesthetic and oncological care. We assessed, for the first time in a large and homogeneous series of *IDH*-wildtype glioblastomas, the positive impact of awake resection on overall survival.

In our cohort, an awake resection was performed in only 42/264 patients (15.9%) who underwent a resection. The patients referred for awake resection were younger, had smaller tumors, less elevated intracranial pressure, fewer focal neurologic deficits, better KPS scores, and better RTOG-RPA classes, which explains the better outcomes in univariate analyses. This suggests that these patients were carefully selected before being eligible for awake resection. This selection partly explains the safety and efficacy of the awake surgery in the present series. However, multivariable analyses confirmed that the prognostic advantage of awake surgery on progression-free and overall survivals were independent from age, tumor volume, clinical signs, KPS score, RTOG-RPA classes, and neurosurgeon experience, and suggested the additional survival benefit of awake surgery together with total or supratotal surgical resection. Similarly, the case-matched analysis confirmed the prognostic advantage of awake resection on progression-free and overall survivals despite individual matching based on sex, age, tumor location, side, and volume, RTOG-RPA classes, and experience of the neurosurgeons. In addition, the survival benefit remained significant in the subgroup of 223 patients operated on by the two neurosurgeons expert both in awake and asleep surgery, which suggested that the neurosurgeon’s level of experience and technical preference parameters cannot account for the observed results.

Of note, 4/42 patients (9.5%) operated on under awake conditions presented with preoperative elevated intracranial pressure, which is normally considered a formal contraindication to awake surgery [19,43]. It is mandatory to control the intracranial pressure and mass effect to achieve accurate intraoperative functional brain mapping since both can mask eloquent brain areas, as previously reported [44,45]. In 6/42 patients (14.3%), we faced intraoperative elevated intracranial pressure due to tumor mass effect. We took advantage of intraoperative mass effect alleviation to unmask and identify eloquent brain areas during awake functional mapping. On a practical basis, our patient selection algorithm allowed for awake resection to be achieved in all cases.

We reported higher resection rates, lower residual tumor rates, and more supratotal resections with awake resections than with asleep resections. This highlights the positive impact of awake brain mapping on the extent of resection of *IDH*-wildtype glioblastomas, which is already considered the gold standard for lower-grade and *IDH*-mutant diffuse gliomas [27]. Similar to previous reports, we found a survival benefit for patients who had total and supratotal resections [20,21,24,39,40,41,46,47]. This suggests that resecting the apparently normal tissue at the tumor periphery decreases the number of remaining infiltrating isolated glioma cells. In a recent literature review, the impact of supratotal resection on overall survival in glioblastoma patients was assessed at an evidence level of III [48]. Li et al. previously reported the largest series of resected glioblastomas, showing that pushing the boundary to 100% resection and beyond, along with the removal of a significant amount of the FLAIR abnormality region, may result in longer survival without significant increases in postoperative morbidity [47]. Molinaro et al. recently reported a series of 478 newly diagnosed *IDH*-wildtype glioblastomas showing that both maximal resection of contrast-enhancing and non-contrast-enhancing parts of the tumor may impact overall survival, regardless of the *MGMT* promoter methylation status [3]. Furthermore, they identified four overall survival risk categories based on age, Temozolomide treatment, contrast-enhancing tumor resection, and preoperative and postoperative non-contrast-enhancing tumor volume [3]. The present series recall the results of Molinaro et al., with improved survival for patients with classes 3–4 than for patients with classes 1–2. In addition, we observed more patients with Molinaro classes 3–4 following awake surgery than following asleep surgery, which suggests that awake surgery allows larger resections beyond the contrast-enhanced part of the tumor.

Despite tumor location within eloquent brain areas and despite aggressive resections, postoperative complications were lower and neurological outcomes were better following awake resections compared to asleep resections. Previous studies have reported that a greater extent of resection was associated with a lower rate of postoperative complications in glioblastomas [41,49,50] and described the correlation between awake craniotomy and function preservation in series of diffuse gliomas located within eloquent brain areas [51,52,53]. Clavreul et al., showed a correlation between incomplete resection and postoperative neurological deficits in a series of glioblastomas resected under awake craniotomy [42]. Nakajima et al. reported that awake surgery preserved long-term independence levels in glioblastoma patients [54]. Zigiotto et al., showed that awake resection can lead to an overall survival benefit while preserving neurological and cognitive functions in glioblastoma patients [23]. The tolerance of awake surgery is illustrated by 100% of patients receiving standard radiochemotherapy, which remains one of the main survival predictors [1]. Altogether, the observed outcomes support the safety, feasibility, and efficacy of awake resections, performed by an experienced glioma neurosurgeon, for supratentorial *IDH*-wildtype glioblastomas in selected patients.

### 4.3. Generalizability

This study controlled for histomolecular biases by selecting a homogeneous population of newly diagnosed supratentorial *IDH*-wildtype glioblastomas with re-assessment of all diffuse gliomas under study according to the 2016 updated WHO classification. We provided the volumetric extent of resection, postoperative outcomes, Molinaro survival risk categories, and survival analyses that are required standards for a reliable evaluation of the safety and efficacy of awake surgery. In addition, we provided a case-matched analysis to control for selection biases between awake resection and asleep resection subgroups, including the neurosurgeon’s level of experience (19 awake patients were matched with patients operated on by the same neurosurgeon expert both in awake and asleep surgery, 17 awake patients were matched with patients operated on by a neurosurgeon expert in asleep surgery, only six awake patients were matched with patients operated on by a general neurosurgeon).

### 4.4. Limitations

The interpretation of the present results should be reviewed with some limitations. The single-institution retrospective design of the study, the exploratory design of statistical analyses with inherent selection and treatment biases, including the fact that the awake surgery was not randomly assigned, limit the generalizability of the results. The potential bias induced by data missing of the *MGMT* promoter methylation status was limited as much as possible by their systematic incorporation in statistical analyses as a specific category. No causal conclusion can be directly made on the effects of awake resection. Further confirmatory studies, possibly with a randomized awake/asleep approach, in a homogenous group of presumably resectable IDH-wildtype glioblastoma patients, performed by experienced neurosurgeons who mastered both awake and asleep surgery, should be proposed to assess the impact of awake surgery on patients’ neurocognitive status, quality of life, extent of resection, and survival.

## 5. Conclusions

Awake surgery is safe, allows for larger resections than asleep surgery, and positively impacts survival in carefully selected *IDH*-wildtype glioblastoma adult patients. In a select number of *IDH*-wildtype glioblastoma patients, awake resection should be proposed as a first-line treatment.

## Figures and Tables

**Figure 1 cancers-13-02911-f001:**
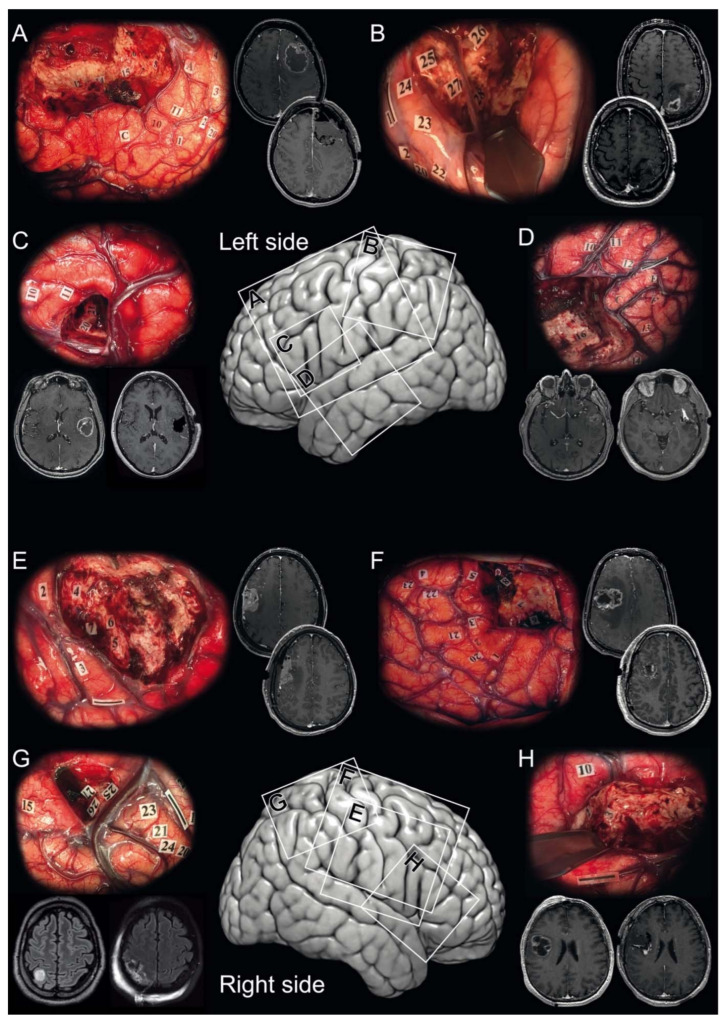
Illustrative cases. Illustrative cases of isocitrate dehydrogenase (*IDH*)-wildtype glioblastoma awake resection using direct cortico-subcortical electric stimulations to define functional boundaries. Intraoperative photographs showing the surgical field with the functional boundaries of the resection marked intraoperatively with numbered tags in the surgical cavity and corresponding pre- and postoperative magnetic resonance post-contrast T1-weighted imaging. (**A**) A 46-year-old right-handed woman underwent a supratotal awake resection (36.9 cm^3^, no residual tumor) for a left frontal *IDH*-wildtype glioblastoma. (Mapping was performed at 2.0 mA. Numbered tags: involuntary movement of the mouth and tongue at 1 and 2, involuntary movement of the hand at 3 and involuntary movement of the elbow at 4 identifying the primary motor cortical pathways; paresthesias of the tongue at 20 identifying the sensory cortical pathways; anarthria at 10 and 11 identifying language cortical pathways; arrest of voluntary movements of the upper limb at 5 and 6 and arrest of voluntary movements of the upper limb and of speech at 7 identifying cortico-subcortical negative motor networks; phonemic paraphasias at 12, 14, and 15 identifying the language subcortical dorsal phonologic pathway). (**B**) A 45-year-old right-handed man underwent a partial awake resection (62.5 cm^3^, 19.7 cm^3^ of residual tumor) for a left parietal *IDH*-wildtype glioblastoma. (Mapping was performed at 5.0 mA. Numbered tags: involuntary movement of the hand occurred at 1 and 2 identifying the primary motor cortical pathways; paresthesias of the hand at 20 and 22 and paresthesias of the shoulder at 23 and 24 identifying the sensory cortical pathways; paresthesias of the lower back at 25 and 27 and paresthesias of the lower limb at 26 and 28 identifying the sensory subcortical pathways). (**C**) A 63-year-old right-handed woman underwent a total awake resection (14.6 cm^3^, no residual tumor) for a left frontal *IDH*-wildtype glioblastoma. (Mapping was performed at 5.0 mA. Numbered tags: anarthria occurred at 10 identifying language cortical pathways; dysarthria at 11 identifying the primary motor cortical pathways; phonemic paraphasias at 20 and 21 identifying the language subcortical dorsal phonologic pathway). (**D**) A 66-year-old right-handed woman underwent a total awake resection (19.6 cm^3^, no residual tumor) for a left temporal *IDH*-wildtype glioblastoma. (Mapping was performed at 5.0 mA. Numbered tags: anarthria occurred at 10, and semantic paraphasias at 13 and 14 identifying language cortical pathways; dysarthria at 11 and 12 identifying the primary motor cortical pathways; latency at 15 and 17 and semantic paraphasia at 16 identifying the language subcortical ventral semantic pathway). (**E**) A 44-year-old right-handed woman underwent a subtotal awake resection (18.9 cm^3^, 0.6 cm^3^ of residual tumor) for a right frontal *IDH*-wildtype glioblastoma. (Mapping was performed at 3.5 mA. Numbered tags: arrest of voluntary movements of the upper limb occurred at 2, 3 and 4 identifying cortico-subcortical negative motor networks; involuntary movements of the tongue at 5, 6 and 7 identifying subcortical primary motor pathways). (**F**) A 26-year-old left-handed man underwent a supratotal awake resection (37.9 cm^3^, no residual tumor) for a right frontal *IDH*-wildtype glioblastoma. (Mapping was performed at 2.0 mA. Numbered tags: involuntary movement of the jaw and tongue occurred at 1, 2 and 3, involuntary movement of the hand at 4, and involuntary movement of the wrist at 5 identifying the primary motor cortical pathways; paresthesias of the lips at 20 and 21, paresthesias of the thumb at 22, paresthesias of third, fourth, fifth fingers at 23 identifying the sensory cortical pathways; involuntary movements of the jaw at 7 identifying subcortical primary motor pathways; arrest of voluntary movements of the mouth at 6 and saccadic lateral deviation of the eyes at 8 identifying cortico-subcortical negative motor networks). (**G**) A 54-year-old left-handed man underwent a total awake resection (2.4 cm^3^, no residual tumor) for a right parietal *IDH*-wildtype glioblastoma. (Mapping was performed at 3.0 mA. Numbered tags: involuntary movement of the wrist occurred at 1 and 2 identifying the primary motor cortical pathways; paresthesias of thumb at 20, paresthesias of second and third fingers at 24, paresthesias of fourth and fifth fingers at 21 and 23 identifying the sensory cortical pathways; latency during picture naming at 15; paresthesias of the thorax at 25, paresthesias of the upper limb at 27, and paresthesias of the lower limb at 27 identifying the sensory subcortical pathways). (**H**) A 54-year-old right-handed woman underwent a supratotal awake resection (28.9 cm^3^, no residual tumor) for a right frontal *IDH*-wildtype glioblastoma. (Mapping was performed at 3.5 mA. Numbered tags: dysarthria occurred at 10 identifying the primary motor cortical pathways; involuntary movements of the jaw at 2 and of the lips at 3 identifying subcortical primary motor pathways).

**Figure 2 cancers-13-02911-f002:**
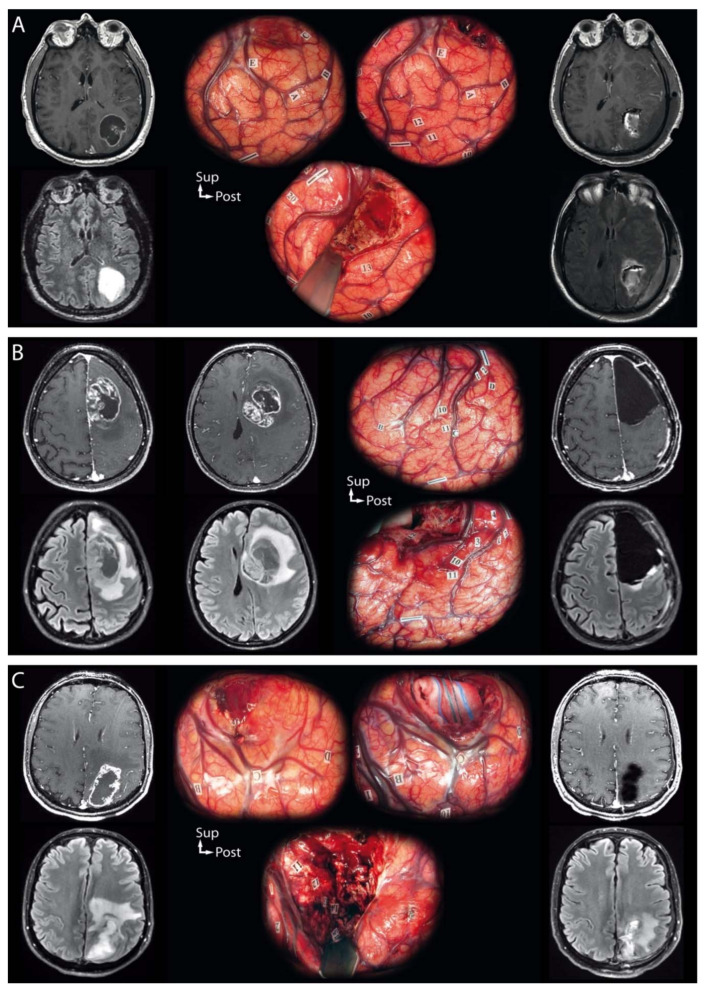
Illustrative cases of the intraoperative management of tumor-related elevated intracranial pressure and local mass effect during awake resection using intraoperative direct cortico-subcortical electrostimulation mapping of *IDH*-wildtype glioblastomas. On the left, preoperative magnetic resonance examinations (post-contrast T1-weighted and Fluid Attenuated Inversion Recovery sequences); in the middle, intraoperative photographs with eloquent sites tagged; on the right, day-one postoperative magnetic resonance examinations (post-contrast T1-weighted and Fluid Attenuated Inversion Recovery sequences). (**A**) A 63-year-old right-handed man presented with uncontrolled focal seizures revealing a left cystic contrast-enhancing necrotic parietal tumor. An awake resection was performed using intraoperative direct cortico-subcortical electrostimulation mapping. The initial cortical mapping (up to 6.0 mA) failed at identifying any eloquent sites. Intralesional debulking was performed, which reduced mass effect. Subsequent cortical mapping allowed for the identification of the primary motor cortex of the face (1) and hand (2), and of latency and semantic paraphasia in the supramarginal gyrus (10, 11, 12, 13). Then, the resection was performed beyond the limits of the solid tumor tissue component according to subcortical functional boundaries, with the arcuate fasciculus as the lateral limit of the surgical cavity (latency, 14, 15, 16). No visual disturbances were observed at the inferior limits of the surgical cavity. (**B**) A 39-year-old right-handed woman presented with signs of elevated intracranial pressure and language impairment revealing a left cystic contrast-enhancing necrotic frontal tumor. An awake resection was performed using intraoperative direct cortico-subcortical electrostimulation mapping. The initial cortical mapping (3.5 mA) allowed for the identification of the ventral premotor cortex inducing speech arrest (10, 11) and of the primary motor cortex of the hand (1, 2) with no other response elicited on cortical mapping. An ultrasound-guided cyst puncture was performed to reduce mass effect, which revealed the cortical negative motor networks inducing arrest of voluntary movements of the hand (3, 4) upon electrostimulation. Then, the resection was performed beyond the limits of the solid tumor tissue component according to subcortical functional boundaries, with the white matter involved in motor control as the posterior limit of the surgical cavity (arrest of voluntary movements, 5, 6, 7) and the arcuate fasciculus as the lateral limit of the surgical cavity (phonemic paraphasia, 12). (**C**) A 43-year-old right-handed man presented with focal seizures revealing a left contrast-enhancing and necrotic parietal tumor. An awake resection was performed using intraoperative direct cortico-subcortical electrostimulation mapping. Upon opening the dura, brain herniation occurred, and the patient experienced headaches, leading to initial cortical mapping failure. Intralesional debulking was performed, which reduced the mass effect and headaches. Subsequent cortical mapping (4.0 mA) allowed for the identification of the primary sensory cortex of the hand (1) and upper limb (4) and of semantic paraphasia in the supramarginal gyrus (10). Then, the resection was performed beyond the limits of the solid tumor tissue component according to subcortical functional boundaries, with the white matter involved in sensory control as the anterior limit of the surgical cavity (11, 12, 13, 14 for the lower limb; 15 for the upper limb). No visual or language disturbances were observed at the lateral and inferior limits of the surgical cavity.

**Figure 3 cancers-13-02911-f003:**
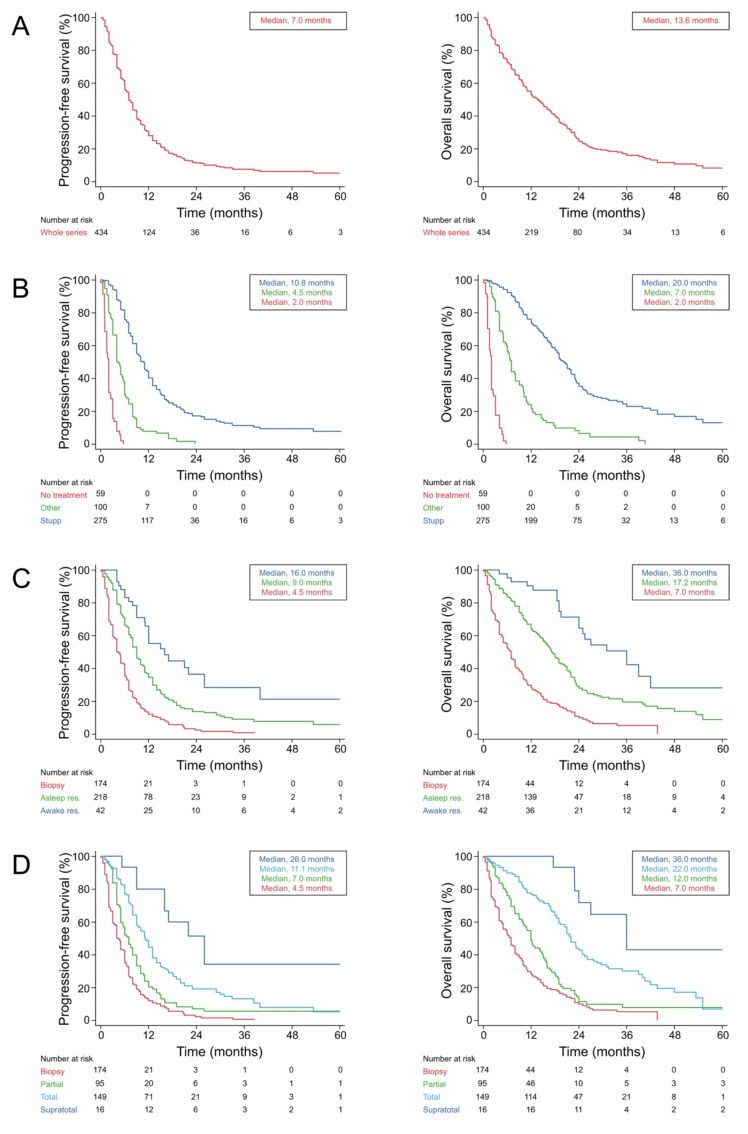
Kaplan–Meier estimates of progression-free and overall survival. (**A**) Kaplan–Meier estimates of progression-free survival (left) and overall survival (right) in the whole study population (434 patients). (**B**) Kaplan–Meier estimates of progression-free survival (left) and overall survival (right) according to the first-line oncological treatment received following surgery: standard radiochemotherapy (Stupp, blue line), other oncological treatment (Other, green line), and no treatment (No treatment, red line). (**C**) Kaplan–Meier estimates of progression-free survival (left) and overall survival (right) according to the type of surgery performed: biopsy (Biopsy, red line), surgical resection under asleep conditions (Asleep, green line), and surgical resection under awake conditions (awake, blue line). (**D**) Kaplan–Meier estimates of progression-free survival (left) and overall survival (right) according to the extent of surgical resection: biopsy (Biopsy, red line), partial surgical resection (Partial, green line), total surgical resection (total, light blue line), and supratotal surgical resection (supratotal, blue line).

**Table 1 cancers-13-02911-t001:** Awake resection and asleep resection subgroups paired by matching criteria.

	Awake Resection Subgroup							Asleep Resection Subgroup						
Pt	Sex	Age	RTOG-RPA Class	Side	Lobe	Volume	Neurosurgeon	Sex	Age	RTOG-RPA Class	Side	Lobe	Volume	Neurosurgeon
Complete match														
3	F	48	3–4	L	F	>	Expert Awake-Asleep	F	49	3–4	L	F	>	Expert Asleep
5	F	63	3–4	L	F	<	Expert Awake-Asleep	F	65	3–4	L	F	<	Expert Awake-Asleep
7	F	72	5–5	L	F	<	Expert Awake-Asleep	F	68	5–5	L	F	<	Expert Awake-Asleep
8	F	50	3–4	R	T	<	Expert Awake-Asleep	F	55	3–4	R	T	<	Expert Asleep
10	F	65	3–4	L	T	<	Expert Awake-Asleep	F	70	3–4	L	T	<	Expert Awake-Asleep
11	F	40	3–4	L	P	<	Expert Awake-Asleep	F	47	3–4	L	P	<	Expert Awake-Asleep
13	M	46	3–4	R	F	<	Expert Awake-Asleep	M	44	3–4	R	F	<	Expert Awake-Asleep
14	M	51	3–4	R	F	<	Expert Awake-Asleep	M	51	3–4	R	F	<	Expert Awake-Asleep
16	M	62	3–4	L	F	<	Expert Awake-Asleep	M	57	3–4	L	F	<	Expert Asleep
17	M	66	5–5	L	F	<	Expert Awake-Asleep	M	63	5–5	L	F	<	Expert Asleep
18	M	68	3–4	R	F	<	Expert Awake-Asleep	M	71	3–4	R	F	<	Expert Awake-Asleep
20	M	47	3–4	L	T	<	Expert Awake-Asleep	M	45	3–4	L	T	<	Expert Asleep
21	M	50	3–4	L	T	>	Expert Awake-Asleep	M	47	3–4	L	T	>	Expert Awake-Asleep
22	M	66	3–4	L	T	<	Expert Awake-Asleep	M	68	3–4	L	T	<	Expert Asleep
24	M	45	3–4	L	P	<	Expert Awake-Asleep	M	46	3–4	L	P	<	Expert Asleep
26	M	44	3–4	R	F	<	Expert Awake-Asleep	M	46	3–4	R	F	<	Expert Awake-Asleep
27	M	62	3–4	L	T	<	Expert Awake-Asleep	M	69	3–4	L	T	<	Expert Awake-Asleep
28	M	59	5–5	L	F	<	Expert Awake-Asleep	M	63	5–5	L	F	<	Expert Asleep
29	M	56	3–4	L	F	<	Expert Awake-Asleep	M	57	3–4	L	F	<	Expert Asleep
31	M	51	5–5	L	T	<	Expert Awake-Asleep	M	53	5–5	L	T	<	Expert Asleep
33	M	55	3–4	L	T	<	Expert Awake-Asleep	M	50	3–4	L	T	<	Expert Asleep
34	M	26	3–4	R	F	>	Expert Awake-Asleep	M	32	3–4	R	F	>	Expert Awake-Asleep
40	M	49	3–4	L	P	<	Expert Awake-Asleep	M	46	3–4	L	P	<	Expert Asleep
41	M	54	3–4	R	P	<	Expert Awake-Asleep	M	56	3–4	R	P	<	Expert Awake-Asleep
Incomplete match in one criterion £														
2	F	46	3–4	R	F	<	Expert Awake-Asleep	F	46	3–4	L	F	<	Expert Awake-Asleep
4	F	54	3–4	L	F	<	Expert Awake-Asleep	F	45	3–4	R	F	<	Expert Asleep
9	F	59	3–4	L	T	<	Expert Awake-Asleep	F	64	3–4	L	T	<	General
19	M	40	3–4	L	T	<	Expert Awake-Asleep	M	40	3–4	L	T	<	General
25	M	66	5–5	L	T	>	Expert Awake-Asleep	M	61	5–5	L	T	<	Expert Asleep
32	M	51	3–4	L	T	<	Expert Awake-Asleep	M	47	3–4	L	T	>	Expert Awake-Asleep
39	M	29	3–4	L	T	<	Expert Awake-Asleep	M	34	3–4	R	T	<	Expert Awake-Asleep
42	M	50	5–5	L	P	>	Expert Awake-Asleep	M	49	5–5	R	P	>	Expert Awake-Asleep
Incomplete match in two criteria £														
1	F	33	3–4	L	T	<	Expert Awake-Asleep	F	40	3–4	R	T	>	Expert Awake-Asleep
6	F	65	5–5	R	F	>	Expert Awake-Asleep	F	58	3–4	R	F	<	Expert Asleep
12	F	61	5–5	L	P	>	Expert Awake-Asleep	F	60	3–4	L	P	<	Expert Asleep
15	M	54	5–5	R	F	<	Expert Awake-Asleep	M	55	3–4	R	F	<	General
23	M	42	3–4	L	P	>	Expert Awake-Asleep	M	42	3–4	B	P	<	Expert Asleep
35	M	63	3–4	L	P	<	Expert Awake-Asleep	M	51	3–4	L	P	>	Expert Awake-Asleep
37	F	57	3–4	L	I	<	Expert Awake-Asleep	F	60	3–4	R	I	>	Expert Awake-Asleep
38	M	49	3–4	L	F	>	Expert Awake-Asleep	M	47	3–4	R	F	>	General
Incomplete match in three criteria £														
30	M	56	5–5	L	F	<	Expert Awake-Asleep	M	55	3–4	R	F	<	General
36	F	39	3–4	L	F	>	Expert Awake-Asleep	F	27	3–4	L	F	<	General

**Table 2 cancers-13-02911-t002:** Patient and tumor characteristics (*n* = 453).

Characteristics	Whole Series	Asleep Resection Subgroup	Awake Resection Subgroup	Biopsy Subgroup	*p*-Value
Patients (%)	453 (100)	222 (49.0)	42 (9.3)	189 (41.7)	
Age ≤60 years (%) >60 years (%) Mean ± SD	177 (39.1) 276 (60.9) 63.0 ± 12.6	94 (42.3) 128 (57.7) 62.1 ± 11.7	30 (71.4) 12 (28.6) 52.6 ± 10.6	53 (28.0) 136 (72.0) 66.4 ± 12.6	<0.0001 <0.0001
Sex Female (%) Male (%)	202 (44.6) 251 (55.4)	107 (48.2) 115 (51.8)	17 (40.5) 25 (59.5)	78 (41.3) 111 (58.7)	0.3162
Time to diagnosis Mean ± SD	1.6 ± 2.0	1.5 ± 1.8	1.7 ± 1.5	1.7 ± 2.2	0.1032
Volume (cm^3^) Mean ± SD	44.8 ± 50.2	52.1 ± 53.6	23.9 ± 29.3	40.8 ± 48.2	<0.0001
Side Right (%) Left (%) Bilateral (%)	189 (41.7) 202 (44.6) 62 (13.7)	124 (55.9) 80 (36.0) 18 (8.1)	9 (21.4) 33 (78.6) 0 (0.0)	56 (29.6) 89 (47.1) 44 (23.3)	<0.0001
Location Frontal (%) Temporal (%) Parietal (%) Insular (%) Occipital (%) Basal Ganglia (%) Limbic (%)	170 (37.5) 137 (30.2) 89 (19.7) 21 (4.6) 17 (3.8) 17 (3.8) 2 (0.4)	65 (29.3) 92 (41.4) 46 (20.7) 7 (3.2) 12 (5.4) 0 (0.0) 0 (0.0)	18 (42.9) 14 (33.3) 9 (21.4) 1 (2.4) 0 (0.0) 0 (0.0) 0 (0.0)	87 (46.0) 31 (16.4) 34 (18.0) 13 (6.9) 5 (2.7) 17 (9.0) 2 (1.0)	<0.0001
Presenting Symptom Asymptomatic (%) Epileptic seizures (%) Elevated intracranial pressure (%) Focal neurologic deficit (%)	8 (1.8) 144 (31.8) 69 (15.2) 232 (51.2)	5 (2.3) 66 (29.7) 46 (20.7) 105 (47.3)	1 (2.4) 27 (64.2) 3 (7.1) 11 (26.2)	2 (1.1) 51 (27.0) 20 (10.6) 116 (61.4)	<0.0001
Elevated intracranial pressure No (%) Yes (%)	315 (69.5) 138 (30.5)	127 (57.2) 95 (42.8)	38 (90.5) 4 (9.5)	150 (79.4) 39 (20.6)	<0.0001
Epileptic seizures at surgery No (%) Yes (%)	281 (62.0) 172 (38.0)	142 (64.0) 80 (36.0)	10 (23.8) 32 (76.2)	129 (68.5) 60 (31.5)	<0.0001
Focal neurologic deficit at surgery No (%) Yes (%)	133 (29.4) 320 (70.6)	69 (31.1) 153 (68.9)	22 (52.4) 20 (47.6)	42 (22.2) 147 (77.8)	0.0006
KPS score >70 (%) ≤70 (%) Mean ± SD	267 (58.9) 186 (41.1) 75.5 ± 15.8	139 (62.6) 83 (37.4) 76.7 ± 4.8	37 (88.1) 5 (11.9) 85.5 ± 10.9	91 (48.2) 98 (51.8) 71.8 ± 16.6	<0.0001 <0.0001
RTOG-RPA Class Class 3-4 (%) Class 5-6 (%)	207 (45.7) 246 (54.3)	141 (63.5) 81 (36.5)	32 (76.2) 10 (23.8)	34 (18.0) 155 (82.0)	<0.0001
MGMT promoter methylation status No (%) Yes (%) Not available (%)	76 (16.8) 60 (13.2) 317 (70.0)	61 (27.5) 40 (18.0) 121 (54.5)	5 (11.9) 7 (16.7) 30 (71.4)	10 (5.3) 13 (6.9) 166 (87.8)	<0.0001
Neurosurgeon Expert both in awake and asleep surgery Expert only in asleep surgery General	223 (49.2) 116 (25.6) 114 (25.2)	89 (40.1) 52 (23.4) 81 (36.5)	42 (100) 0 (0) 0 (0)	92 (48.7) 64 (33.9) 33 (17.5)	<0.0001
Extent of resection Mean resection ± SD Mean residual ± SD Partial (%) Total (%) Supratotal (%)	98 (21.6) 150 (33.1) 16 (3.5)	92.9 ± 15.1 7.3 ± 15.6 89 (40.1) 126 (56.8) 7 (3.1)	93.9 ± 18.7 6.3 ± 19.2 9 (21.4) 24 (57.2) 9 (21.4)		0.0313 0.0306 <0.0001
Surgery-related Complications Surgical site hematoma (%) Surgical site infection (%) Systemic infection (%) Seizures worsening (%) Focal neurologic deficit worsening (%) Thrombosis (%)	10 (2.2) 9 (2.0) 10 (2.2) 20 (4.4) 69 (15.2) 9 (2.0)	3 (1.4) 8 (3.6) 7 (3.2) 11 (5.0) 42 (18.9) 6 (2.7)	1 (2.4) 1 (0.2) 2 (4.8) 0 (0.0) 8 (19.0) 0 (0.0)	6 (3.2) 0 (0.0) 1 (0.5) 9 (4.8) 19 (10.1) 3 (1.6)	0.4490 0.0330 0.0975 0.1358 0.0299 0.4514
One-Month Postoperative Death (%) No (%) Yes (%)	437 (96.5) 16 (3.5)	219 (98.6) 3 (1.4)	42 (100) 0 (0.0)	176 (93.1) 13 (6.9)	0.0025
Postoperative Oncological Treatment Standard Radiochemotherapy Protocol Radiotherapy alone (%) Temozolomide alone (%) Radiotherapy followed by Temozolomide (%) Other chemotherapy Supportive care (%) Lost to Follow-Up (%)	275 (60.7) 45 (9.9) 27 (6.0) 26 (5.7) 2 (0.5) 59 (13.0) 19 (4.2)	165 (74.3) 18 (8.1) 8 (3.6) 15 (6.8) 0 (0.0) 12 (5.4) 4 (1.8)	38 (90.4) 2 (4.8) 0 (0.0) 0 (0.0) 2 (4.8) 0 (0.0) 0 (0.0)	72 (38.1) 25 (13.2) 19 (10.1) 11 (5.8) 0 (0.0) 47 (24.9) 15 (7.9)	<0.0001
Time to Oncological Treatment Weeks ± SD	5.5 ± 2.5	5.9 ± 2.5	4.2 ± 2.5	5.2 ± 2.2	0.0008
Molinaro’s Classes (1-4) 1 2 3 4 Biopsy only (excluded)	7 (1.5) 114 (25.2) 121 (26.7) 22 (4.9) 189 (41.7)	6 (2.7) 104 (46.9) 96 (43.2) 16 (7.2) 0 (0.0)	1 (2.4) 10 (23.8) 25 (59.5) 6 (14.3) 0 (0.0)	0 (0.0) 0 (0.0) 0 (0.0) 0 (0.0) 189 (100)	0.0325
Progression-Free Survival Median (months) [95% CI]	7.0 [6.5–8.0]	9.0 [8.0–10.0]	16.0 [11.1–26.0]	4.7 [4.0–6.0]	<0.0001
Overall Survival Median (months) [95% CI]	13.6 [12.0–16.0]	17.2 [15.1–19.4]	36.0 [24.0–42.0]	7.0 [5.0–8.0]	<0.0001

MGMT: O6-methylguanine-DNA methyltransferase; KPS: Karnofsky Performance Status; RTOG: Radiation Therapy Oncology Group; RPA: Recursive Partitioning Analysis; SD: Standard Deviation.

**Table 3 cancers-13-02911-t003:** Univariate and multivariate predictors of progression-free survival and overall survival in the whole series (*n* = 434). Unadjusted hazard ratios by log-rank tests and adjusted hazard ratios by Cox proportional hazards model.

Parameter	Progression-Free Survival						Overall Survival					
	Unadjusted Hazard Ratio			Adjusted Hazard Ratio			Unadjusted Hazard Ratio			Adjusted Hazard Ratio		
	uHR	CI95%	*p*-Value	aHR	CI95%	*p*-Value	uHR	CI95%	*p*-Value	aHR	CI95%	*p*-Value
Age ≤60 years >60 years	1 (ref) 1.25	0.99–1.58	0.0638				1 (ref) 1.76	1.40–2.21	<0.0001			
Sex Female Male	1 (ref) 1.07	0.87–1.31	0.4992				1 (ref) 1.06	0.76–1.32	0.6101			
Volume ≤28 cm^3^ >28 cm^3^	1 (ref) 1.13	0.92–1.38	0.2305				1 (ref) 1.34	1.08–1.67	0.0074			
Location Frontal Temporal Parietal Insular Occipital Deep seated	1 (ref) 0.95 0.76 1.41 0.82 2.12	0.72–1.25 0.54–1.08 0.77–2.57 0.44–1.54 0.97–4.61	0.7175 0.1246 0.2640 0.5437 0.0586				1 (ref) 0.79 0.80 2.07 0.57 2.29	0.61–1.03 0.59–1.10 1.27–3.37 0.30–1.04 1.34–3.94	0.0793 0.1763 0.0033 0.0687 0.0025			
KPS score ≤70 >70	1 (ref) 0.56	0.45–0.69	<0.0001				1 (ref) 0.49	0.39–0.61	<0.0001	1 (ref) 0.66	0.52–0.85	0.0013
RTOG-RPA classes * 3-4 5-6	1 (ref) 1.43	1.13–1.82	0.0031				1 (ref) 3.43	2.13–5.52	<0.0001			
MGMT promoter Non methylated Methylated Not available	1 (ref) 0.64 1.01	0.45–0.92 0.78–1.32	0.0159 0.9109				1 (ref) 0.54 1.01	0.36–0.79 0.77–1.32	0.0017 0.9293	1 (ref) 0.55 1.04	0.37–0.82 0.48–1.87	0.0031 0.4043
Neurosurgeon General Expert in Glioma	1 (ref) 0.71	0.42–1.19	0.1946				1 (ref) 0.89	0.71–1.12	0.3176			
Extent of resection Biopsy Supratotal Total Partial	1 (ref) 0.17 0.38 0.62	0.08–0.33 0.29–0.48 0.48–0.81	<0.0001 <0.0001 0.0005	1 (ref) 0.31 0.52 0.81	015–0.65 0.40–0.68 0.61–1.06	0.0019 <0.0001 0.13	1 (ref) 0.13 0.30 0.63	0.06–0.29 0.23–0.39 0.48–0.83	<0.0001 <0.0001 0.0013	1 (ref) 0.27 0.43 0.72	0.12–0.62 0.32–0.57 0.54–0.98	0.0021 <0.0001 0.0366
Surgery Asleep Awake	1 (ref) 0.39	0.26–0.59	<0.0001	1 (ref) 0.61	0.40–0.93	0.0157	1 (ref) 0.33	0.21–0.52	<0.0001	1 (ref) 0.54	0.33–0.89	0.0156
Treatment Abstention Other treatment Stupp protocol	1 (ref) 0.19 0.06	0.13–0.28 0.04–0.09	<0.0001 <0.0001	1 (ref) 0.21 0.08	0.14–0.30 0.05–0.12	<0.0001 <0.0001	1 (ref) 0.38 0.16	0.22–0.67 0.12–0.21	0.0006 <0.0001	1 (ref) 0.39 0.22	0.22–0.68 0.16–0.29	0.0010 <0.0001
Molinaro’s classes * Group 1 Group 2 Group 3 Group 4	1 (ref) 0.15 0.11 0.12	0.04–0.63 0.02–0.47 0.03–0.57	0.0097 0.0030 0.0072				1 (ref) 0.12 0.06 0.05	0.05–0.31 0.02–0.16 0.02–0.16	<0.0001 <0.0001 <0.0001			

HR: Hazard Ratio; MGMT: O6-methylguanine-DNA methyltransferase; KPS: Karnofsky Performance Status; RTOG: Radiation Therapy Oncology Group; RPA: Recursive Partitioning Analysis. * Not entered in multivariable analyses.

**Table 4 cancers-13-02911-t004:** Univariate and multivariate predictors of progression-free survival and overall survival in the subgroup of patients operated on by neurosurgeon expert both in awake and asleep surgery (*n* = 223). Unadjusted hazard ratios by logrank tests and adjusted hazard ratios by Cox proportional hazards model.

Parameter	Progression-Free Survival						Overall Survival					
	Unadjusted Hazard Ratio			Adjusted Hazard Ratio			Unadjusted Hazard Ratio			Adjusted Hazard Ratio		
	uHR	CI95%	*p*-Value	aHR	CI95%	*p*-Value	uHR	CI95%	*p*-Value	aHR	CI95%	*p*-Value
Age ≤60 years >60 years	1 (ref) 1.12	0.79–1.56	0.5228				1 (ref) 1.92	1.40–2.65	<0.0001			
Sex Female Male	1 (ref) 1.05	0.75–1.48	0.7716				1 (ref) 1.01	0.73–1.37	0.9856			
Volume ≤ 28 cm^3^ > 28 cm^3^	1 (ref) 1.17	0.95–1.54	0.4556				1 (ref 1.28	1.02–1.54	0.0138			
Location Frontal Temporal Parietal Insular Occipital Deep seated	1 (ref) 1.03 0.73 2.12 0.88 2.77	0.70–1.53 0.44–1.22 0.95–4.69 0.39–1.95 0.85–9.02	0.8586 0.2351 0.0648 0.7545 0.0911				1 (ref) 0.79 0.73 2.10 0.39 3.55	0.56–1.15 0.45–1.17 1.04–4.25 0.15–0.99 1.51–8.33	0.2285 0.1879 0.0390 0.0479 0.0036			
KPS score ≤70 >70	1 (ref) 0.59	0.39–0.86	0.0067				1 (ref) 0.41	0.29–0.56	<0.0001	1 (ref) 0.51	0.35–0.74	0.0003
RTOG-RPA classes * 3-4 5-6	1 (ref) 1.46	1.03–2.06	0.0321				1 (ref) 2.79	2.03–3.84	<0.0001			
MGMT promoter Non methylated Methylated Not available	1 (ref) 0.43 0.72	0.24–0.74 0.49–1.07	0.0027 0.1080	1 (ref) 0.46 0.62	0.27–0.81 0.41–1.26	0.0073 0.2820	1 (ref) 0.42 0.97	0.24–0.74 0.66–1.40	0.0027 0.8599	1 (ref) 0.42 1.18	0.23–0.75 0.78–1.60	0.0037 0.5422
Extent of resection Biopsy Supratotal Total Partial	1 (ref) 0.26 0.53 0.83	0.12–0.53 0.36–0.78 0.49–1.37	0.0002 0.0012 0.4646	1 (ref) 0.35 0.62 0.81	0.15–0.81 0.39–0.98 0.46–1.43	0.0145 0.0433 0.4698	1 (ref) 0.14 0.30 0.56	0.06–0.30 0.21–0.43 0.35–0.89	<0.0001 <0.0001 0.0142	1 (ref) 0.31 0.46 0.67	0.13–0.76 0.29–0.71 0.39–1.14	0.0098 0.0005 0.1375
Surgery Asleep Awake	1 (ref) 0.50	0.32–0.78	0.0019	1 (ref) 0.63	0.39–0.98	0.0397	1 (ref) 0.34	0.21–0.54	<0.0001	1 (ref) 0.50	0.29–0.85	0.0115
Treatment Abstention Other Treatment Stupp protocol	1 (ref) 0.57 0.23	0.25–1.29 0.14–0.39	0.1792 <0.0001	1 (ref) 0.57 0.25	0.25–1.29 0.15–0.41	0.1973 <0.0001	1 (ref) 0.41 0.16	0.19–0.86 0.11–0.22	0.0184 <0.0001	1 (ref) 0.53 0.23	0.24–1.55 0.15–0.34	0.1099 <0.0001
Molinaro’s classes * Group 1 Group 2 Group 3 Group 4	1 (ref) 0.08 0.06 0.08	0.02–0.39 0.01–0.27 0.02–0.40	0.0016 0.0003 0.0024				1 (ref) 0.12 0.06 0.07	0.03–0.43 0.02–0.21 0.02–0.28	0.0011 <0.0001 0.0002			

HR: Hazard Ratio; MGMT: O6-methylguanine-DNA methyltransferase; KPS: Karnofsky Performance Status; RTOG: Radiation Therapy Oncology Group; RPA: Recursive Partitioning Analysis. * Not entered in multivariable analyses.

## Data Availability

Data is contained within the article.

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
