# Peer review of "Feasibility, Safety and Impact on Overall Survival of Awake Resection for Newly Diagnosed Supratentorial IDH-Wildtype Glioblastomas in Adults"

_cancers, 2021, doi:10.3390/cancers13122911_

Round 1
Reviewer 1 Report
The authors of the paper investigate the role of awake surgery on improving survival in IDH-wildtype glioblastoma patients. They retrospectively investigate a large homogeneous single-institution cohort of 453 newly diagnosed supratentorial IDH-wildtype glioblastoma in adult patients with the goal to assess feasibility, safety and efficacy of awake surgery using univariate, multivariate and case-matched analysis. They found that awake surgery was associated with higher resection rates, lower residual tumor rates, and more supratotal resections than asleep resections, allowed radio-chemotherapy with a shorter time interval in between surgery and radiotherapy. In addition, awake surgery was an independent predictor of progression-free survival and overall survival.
In general, the paper is one of few that focus on the role of awake surgery in high-grade gliomas, and the largest one with a specific focus on IDH-wildtype glioblastoma. Data capture, processing and statistical analysis are done on an adequate manner, and results are presented in a structured way. However, some of the conclusions are overstreched, and the paper suffers of weaknesses that should be clearly pointed out. In addition, the reviewer would suggest to focus more on the most important results and to put others in the appendices, to avoid overload of the paper.
The introduction should be extended with more feasibility / survival data shown on the different modalities of resection, e.g. asleep without monitoring, asleep with monitoring, awake with electrophysiology monitoring, awake with intraoperative testing, awake-awake-awake vs. asleep-awake-asleep, 5-ALA / fluorescein vs. not a.s.o.. Therefore, parts of the discussion section which is pretty extensive could be transferred to the introduction section.
A large subset of patients was biopsied only (about 44%). This is larger than in comparable published series. They write that this was based on the assumption that a safe resection was not possible, however, the proportion remains high, even in an unselected analysis. Could the authors extend on the reasons for this large proportion of biopsy patients?
A large proportion of about 40% of patients got no standard radio-chemotherapy. This is unusually high. The authors should extend on the reasons. In addition, what is meant by “other adjuvant therapy”? Does this mean treatment within clinical trials, and if yes, in what percentage was radiotherapy included in these clinical trials? I would suggest to show radiotherapy treatment in separate, including radiotherapy within clinical trials.
The illustrative cases are indeed instructive. I would however recommend to transfer them to the attachment section, as they provide no scientific data.
The observation that local infections were more prevalent in the asleep group argues for a more careful procedure in the awake subgroup. I cannot see another possible reason for the lower rate, also specifically considering that awake surgery generally takes longer tan asleep surgery. Could the authors elaborate on that?
The progression-free survival and overall survival rates were highly distinct in awake vs. asleep vs. biopsy. This is, in the univariate non-corrected analysis, certainly an effect of favorable prognostic factors in the prognostically better groups. This should be clearly stated. The effects however did hold after correcting with Cox regression analysis and while matching cases. As the statistical methods were correctly applied, this seems to be a true statistical and clinically meaningful effect. In addition, Kaplan Meier estimates for progression-free survival and overall survival are impressive and show a clear dose-effect relation, that makes the dataset plausible. Median times associate with published data, and prognostic factors well relate to prognostic factors in other publications.
The matched pairs analysis comes down to a small number of patients, so the statistical power may be restricted. I cannot comment on that as a non-statistician, but maybe the authors could elaborate on that in more extent.
The tables are difficult to read and to interpret due to their formatting. This should be adjusted.
The authors draw several conclusions that appear to stretched in some cases. The respective conclusions should be toned down, specifically the conclusions that awake surgery:
1) was feasible in select patients with complication rates and neurological deficits inferior or similar to those of asleep resections – that can only be concluded under the vision that these patients were severely preselected;
2) was associated with higher resection rates, lower residual tumor rates, and more supratotal resections than asleep resections – this is certainly only true because patients were preselected for supratotal awake surgery based on MRI and clinical status, while others were pre-planned for biopsy due to tumor location; this should be clearly stated;
3) allowed standard radio-chemotherapy to be performed systematically within a short time between surgery and radiotherapy – why should awake surgery allow for this? This seems to be another preselection bias for fitter patients;
In summary, most of the conclusions depend of preselection factors, which should be clearly outlined in the discussion section.
A general criticism could be that the paper probably more shows the effect of maximal / supramaximal resection and not as much the effect of awake surgery; supratotal resection can also be achieved with asleep surgery, so there is the true value of this approach? This was not specifically investigated in the paper, and the authors should elaborate on that.
The authors themselves name some additional limits of the study, including the retrospective singe-institution design and others. In general, a design with a randomized awake / asleep approach in a homogenous group of presumably resectable patients, performed by experienced neurosurgeons who master both awake and asleep surgery, would be the best way to investigate the true value of the approach. The authors should probably discuss that at the end of their work.
Author Response
Reviewer 1.
The authors of the paper investigate the role of awake surgery on improving survival in IDH-wildtype glioblastoma patients. They retrospectively investigate a large homogeneous single-institution cohort of 453 newly diagnosed supratentorial IDH-wildtype glioblastoma in adult patients with the goal to assess feasibility, safety and efficacy of awake surgery using univariate, multivariate and case-matched analysis. They found that awake surgery was associated with higher resection rates, lower residual tumor rates, and more supratotal resections than asleep resections, allowed radio-chemotherapy with a shorter time interval in between surgery and radiotherapy. In addition, awake surgery was an independent predictor of progression-free survival and overall survival. In general, the paper is one of few that focus on the role of awake surgery in high-grade gliomas, and the largest one with a specific focus on IDH-wildtype glioblastoma. Data capture, processing and statistical analysis are done on an adequate manner, and results are presented in a structured way.
We thank this reviewer for his positive comments.
However, some of the conclusions are overstreched, and the paper suffers of weaknesses that should be clearly pointed out. In addition, the reviewer would suggest to focus more on the most important results and to put others in the appendices, to avoid overload of the paper.
In accordance, we have modified the manuscript according to reviewers’ comments. We present a revised an improved manuscript.
The introduction should be extended with more feasibility / survival data shown on the different modalities of resection, e.g. asleep without monitoring, asleep with monitoring, awake with electrophysiology monitoring, awake with intraoperative testing, awake-awake-awake vs. asleep-awake-asleep, 5-ALA / fluorescein vs. not a.s.o.. Therefore, parts of the discussion section which is pretty extensive could be transferred to the introduction section.
In accordance with this comment, we have integrated several studies showing the impact on the extent of resection of the main intraoperative techniques in the Introduction part, as follows (lines 61-70) : “To improve the extent of resection, several intraoperative techniques have been proposed. 5-ALA fluorescence has been proven effective for improving the overall survival by a randomized multicenter clinical trial(10,14). Sodium fluorescein fluorescence also showed good results in terms of extent of resection but no significant results on overall survival have been reported(15). Similarly, intraoperative MRI showed is impact on glioblastoma’s resection and patient’s survival(16), while intraoperative ultrasound was found useful to perform more radical surgeries and prevent neurological impairment but large series reporting a benefit on survival are still lacking(17,18). To improve the safety of the surgical resection and to improve the benefit-to-risk ratio, awake surgery is the benchmark intraoperative technique for gliomas in eloquent brain areas(19–21)”.
We respectfully believe that the description of specific data regarding awake surgery, which need to be compared with the present results, are best placed in the Discussion section of this revised manuscript, in order to facilitate the reading and the linearity of the paper.
A large subset of patients was biopsied only (about 44%). This is larger than in comparable published series. They write that this was based on the assumption that a safe resection was not possible, however, the proportion remains high, even in an unselected analysis. Could the authors extend on the reasons for this large proportion of biopsy patients?
We confirm that 41.7% of patients under study received a biopsy only. The first reason is that, in our center, we prefer offering a stereotactic biopsy rather than an open biopsy, which may account for a higher count of “biopsy” since open biopsy can be counted as “partial resection”. Of note, the rate >40% of biopsy is within the range of what is observed in our country for the management of glioblastomas in adults. In the only study analyzing the pattern of care at a nationwide level, the reported rate of biopsy is 44% for 952 newly diagnosed meningiomas in adults (Bauchet et al., NeuroOncology 2010).
A large proportion of about 40% of patients got no standard radio-chemotherapy. This is unusually high. The authors should extend on the reasons. In addition, what is meant by “other adjuvant therapy”? Does this mean treatment within clinical trials, and if yes, in what percentage was radiotherapy included in these clinical trials? I would suggest to show radiotherapy treatment in separate, including radiotherapy within clinical trials.
We confirm that patients included in a particular clinical trial were excluded of the present analysis. We have added this information in the revised Material and Methods part, as follow (lines 96-97) : “6) no inclusion in a clinical trial to exclude any particular therapy…”.
In accordance with this comment, we have detailed the “other adjuvant therapy” in the revised Table 2 as follows:
|
Postoperative Oncological Treatment Standard Radiochemotherapy Protocol Radiotherapy alone (%) Temozolomide alone (%) Radiotherapy followed by Temozolomide (%) Other chemotherapy Supportive care (%) Lost to Follow-Up (%) |
275 (60.7) 45 (9.9) 27 (6.0) 26 (5.7) 2 (0.5) 59 (13.0) 19 (4.2) |
165 (74.3) 18 (8.1) 8 (3.6) 15 (6.8) 0 (0.0) 12 (5.4) 4 (1.8) |
38 (90.4) 2 (4.8) 0 (0.0) 0 (0.0) 2 (4.8) 0 (0.0) 0 (0.0) |
72 (38.1) 25 (13.2) 19 (10.1) 11 (5.8) 0 (0.0) 47 (24.9) 15 (7.9) |
|
<0.0001 |
The illustrative cases are indeed instructive. I would however recommend to transfer them to the attachment section, as they provide no scientific data.
We respectfully think that the illustrative cases highlight the message of the present study and provide actual examples of what it feasible for IDH-wildtype glioblastomas located within eloquent brain areas.
The observation that local infections were more prevalent in the asleep group argues for a more careful procedure in the awake subgroup. I cannot see another possible reason for the lower rate, also specifically considering that awake surgery generally takes longer than asleep surgery. Could the authors elaborate on that?
We confirm the lower rate of postoperative infections in the awake subgroup than in the asleep group. To our opinion, the surgical technique (awake or asleep) does not impact the infection rates. The observed discrepancy is possibly due to the younger age and to the better clinical condition of patients of the awake group, and to the surgical experience of neurosurgeons who performed awake surgeries. This is the reason why we have just reported these data without investigating the underlying causes in multivariable and matched pairs analyses.
The progression-free survival and overall survival rates were highly distinct in awake vs. asleep vs. biopsy. This is, in the univariate non-corrected analysis, certainly an effect of favorable prognostic factors in the prognostically better groups. This should be clearly stated. The effects however did hold after correcting with Cox regression analysis and while matching cases. As the statistical methods were correctly applied, this seems to be a true statistical and clinically meaningful effect. In addition, Kaplan Meier estimates for progression-free survival and overall survival are impressive and show a clear dose-effect relation, that makes the dataset plausible. Median times associate with published data, and prognostic factors well relate to prognostic factors in other publications.
We thank this reviewer for his positive comments.
In accordance with this comment, we have rephrased the Discussion part of the revised manuscript as follows (lines 489-498): “The patients referred for awake resection were younger, had smaller tumors, less elevated intracranial pressure, fewer focal neurologic deficits, better KPS scores, and better RTOG-RPA classes, which explains the better outcomes in univariate analyses. This suggests that these patients were carefully selected before being eligible for awake resection. This selection partly explains the safety and efficacy of the awake surgery in the present series. However, multivariable analyses confirmed that the prognostic advantage of awake surgery on progression-free and overall survivals were independent from age, tumor volume, clinical signs, KPS score, RTOG-RPA classes, and neurosurgeon’s experience, and suggested the additional survival benefit of awake surgery together with total or supratotal surgical resection”.
The matched pairs analysis comes down to a small number of patients, so the statistical power may be restricted. I cannot comment on that as a non-statistician, but maybe the authors could elaborate on that in more extent.
The complimentary matched pairs analysis we performed was another way to put to the test the hypothesis that awake surgery positively impacts overall survival. This is just another way to analyze the data. The lower number of patients is counterbalanced by the homogeneous population thanks to data matching. The limitations and biases of the present study, including the statistical ones, are acknowledged in the Limitation part of the Discussion.
The tables are difficult to read and to interpret due to their formatting. This should be adjusted.
In accordance, we have adjusted the format of the Tables to adapt it to the data and to improve readability.
The authors draw several conclusions that appear to stretched in some cases. The respective conclusions should be toned down, specifically the conclusions that awake surgery:
1) was feasible in select patients with complication rates and neurological deficits inferior or similar to those of asleep resections – that can only be concluded under the vision that these patients were severely preselected;
In accordance, we have rephrased this sentence in the revised Discussion part as follows (lines 449-451): “1) was feasible in highly selected patients with complications rates and neurological deficits inferior or similar to those of asleep resections”.
2) was associated with higher resection rates, lower residual tumor rates, and more supratotal resections than asleep resections – this is certainly only true because patients were preselected for supratotal awake surgery based on MRI and clinical status, while others were pre-planned for biopsy due to tumor location; this should be clearly stated;
We confirm that we have acknowledged this point in the Discussion part as follows (lines 491-493): “The patients referred for awake resection were younger, had smaller tumors, less elevated intracranial pressure, fewer focal neurologic deficits, better KPS scores, and better RTOG-RPA classes”.
3) allowed standard radio-chemotherapy to be performed systematically within a short time between surgery and radiotherapy – why should awake surgery allow for this? This seems to be another preselection bias for fitter patients. In summary, most of the conclusions depend of preselection factors, which should be clearly outlined in the discussion section.
Here it is just a fact. We have detailed this information in the “Key points” to underline that awake surgery does not compromise the completion of the standard radio-chemotherapy. In accordance, we have rephrased this sentence in the revised Discussion part as follows (lines): “3) allowed standard radiochemotherapy to be performed systematically without increasing the time interval between surgery and radiotherapy”.
A general criticism could be that the paper probably more shows the effect of maximal / supramaximal resection and not as much the effect of awake surgery; supratotal resection can also be achieved with asleep surgery, so there is the true value of this approach? This was not specifically investigated in the paper, and the authors should elaborate on that.
We confirm that the awake procedure possibly increases the probability to perform a large resection, including a supratotal one. After multiple adjustments using Cox models, both the “extent of resection” and the “awake surgery” parameters independently impacts progression-free and overall survivals.
Of course, the goal is to perform a maximal safe resection and not to perform an awake surgery. Awake surgery is an adjunct to achieve a maximal safe resection.
The authors themselves name some additional limits of the study, including the retrospective singe-institution design and others. In general, a design with a randomized awake / asleep approach in a homogenous group of presumably resectable patients, performed by experienced neurosurgeons who master both awake and asleep surgery, would be the best way to investigate the true value of the approach. The authors should probably discuss that at the end of their work.
We have acknowledged the limitations of the study design in the Limitations part of the manuscript. In accordance with this comment, we have discussed the need of confirmatory studies, ideally with a randomized design in the revised Limitations part as follows (lines 579-584): “No causal conclusion can be directly made on the effects of awake resection. Further confirmatory studies, possibly with a randomized awake/asleep approach, in a homogenous group of presumably resectable IDH-wildtype glioblastoma patients, per-formed by experienced neurosurgeons who master both awake and asleep surgery, should be proposed to assess the impact of awake surgery on patients’ neurocognitive status, quality of life, extent of resection, and survival”.

Reviewer 2 Report
Review summary:
In this article titled “Feasibility, safety and impact on overall survival of awake resection for newly diagnosed supratentorial IDH-wildtype glioblastomas in adults” authors have performed an observational comparative analysis of patients who underwent awake surgery (n=42) with patients who underwent resection under GA (n=222) or Stereotactic biopsy (n=189). Awake surgery was associated with a higher chance of supratotal tumor resection and was also an independent predictor of overall survival in this study population.
I have the following concerns:
- Neurosurgeon experience in awake vs GA does not make sense if one is not attempting that procedure under awake. Obviously, those who are not experienced in awake surgery will not do awake surgery and will be biased towards GA.
- What was the cause of deaths in patients who underwent stereotactic biopsy? If not procedure -related then it’s not a good idea to put that in this table.
- The paper is haphazardly arranged. Please follow the standard guidelines: Introduction, methods, results, discussion, tables, and figures.
- The criteria to choose patients for Awake surgery was based on the surgeon’s preference. This group had a smaller tumor volume which increases the chances of supratotal resection. I believe the authors should add that Awake surgical resection is safe and effective in “carefully selected patients”.
Author Response
Reviewer 2.
In this article titled “Feasibility, safety and impact on overall survival of awake resection for newly diagnosed supratentorial IDH-wildtype glioblastomas in adults” authors have performed an observational comparative analysis of patients who underwent awake surgery (n=42) with patients who underwent resection under GA (n=222) or Stereotactic biopsy (n=189). Awake surgery was associated with a higher chance of supratotal tumor resection and was also an independent predictor of overall survival in this study population.
I have the following concerns:
- Neurosurgeon experience in awake vs GA does not make sense if one is not attempting that procedure under awake. Obviously, those who are not experienced in awake surgery will not do awake surgery and will be biased towards GA.
In the present study, the neurosurgeon’s experience is a possible bias. In accordance, we have incorporated it in all statistical analyses to control for it. Our objective was to perform different analysis to reduce and compensate for selection biases, which are intrinsic in such retrospective design. The present results are consistent with our hypothesis even if the selection bias cannot be completely corrected in this retrospective design.
To mitigate this bias, we have performed : 1) a complimentary analysis in the subgroup of patients operated only by the two neurosurgeons who performed and mastered both awake and asleep surgeries; 2) a survival analysis based on survival groups described by Molinaro et al., and; 3) a case-matched analysis based on the following criteria: sex, age (within 10 years), RTOG-RPA class (3-4 versus 5-6), tumor side, tumor location (same lobe), preoperative volume (cutoff by median), neurosurgeon (expert trained for both awake and asleep surgery versus expert trained only in asleep surgery versus general neurosurgeon).
In the case matched analysis, concerning the neurosurgeon’s experience, 42/42 patients of the awake group were operated on by a neurosurgeon expert both in awake and asleep surgeries. The case matching allowed to obtain 19/42 patients of the asleep group operated on by a neurosurgeon expert both in awake and asleep surgeries, 17/42 patients of the asleep group operated on by a neurosurgeon expert in asleep surgery, and only 6/42 patients operated on by a general neurosurgeon (less than one glioma patient per week).
- What was the cause of deaths in patients who underwent stereotactic biopsy? If not procedure -related then it’s not a good idea to put that in this table.
In accordance, we have removed the data of the 4 “in hospital deaths”. As a precision, we observed 4 early postoperative deaths related to raised intracranial pressure following biopsy. They are still detailed in the “One-month postoperative death”.
- The paper is haphazardly arranged. Please follow the standard guidelines: Introduction, methods, results, discussion, tables, and figures.
The revised manuscript has been re-organized according to standard guidelines.
- The criteria to choose patients for Awake surgery was based on the surgeon’s preference. This group had a smaller tumor volume which increases the chances of supratotal resection. I believe the authors should add that Awake surgical resection is safe and effective in “carefully selected patients”.
In accordance, we have added this information in the revised Conclusion part, as follows (lines 585-588): “Awake surgery is safe, allows larger resections than asleep surgery, and positively impacts survival in carefully selected IDH-wildtype glioblastoma adult patients. In a select number of IDH-wildtype glioblastoma patients, awake resection should be pro-posed as a first-line treatment”.

Round 2
Reviewer 1 Report
In their response letter and the revised manuscript, the authors have addressed all relevant points raised by the reviewer. Some study-intrinsic problems remain, but these have been addressed in the results and discussion sections, especially in the weak points-part of the discussion. The manuscript therefore is fit form publication.